# Determining Residual Deviation and Analysis of the Current Use of the Magnetic Compass

Andrej Androjna [1,*] , Blagovest Belev [2] , Ivica Pavic [3] and Marko Perkovič [1]

1  Faculty of Maritime Studies and Transport, University of Ljubljana, 6320 Portorož, Slovenia; marko.perkovic@fpp.uni-lj.si
2  Nikola Vaptsarov Naval Academy, Black Sea University, 9002 Varna, Bulgaria; bl.belev@naval-acad.bg
3  Faculty of Maritime Studies, University of Split, 21000 Split, Croatia; ipavic71@pfst.hr
*  Correspondence: andrej.androjna@fpp.uni-lj.si

**Abstract:** The use of electronic compasses and satellite systems has led to the magnetic compass becoming a secondary means of navigation. Yet this means of navigating is not only not obsolete, it is a necessary backup device: the construction simplicity of the magnetic compass, without electrical windings, rotating elements, and control units, remains resistant to power losses, hardware malfunction, and thus is reliable under the harshest conditions. This durability, however, comes at some cost; the magnetic compass is influenced by ships' permanent and transient magnetism, cargo gears. For the proper use of a magnetic compass, it is necessary to perform an adjustment to determine the residual deviation at regular intervals. The paper analyses selected methods to manage this, and to identify the main features of classical methods. The research was supplemented by a study carried out during the practical compensations of the magnetic compass at sea and by a survey among navigation officers on its basic requirements for proper use. The results indicate insufficient inspection of the magnetic compass. Further, an investigation into the causes of deviation delivers information regarding the causes under varying conditions including type of ship and latitudinal circumstances. This paper presents findings and recommendations to improve the compensation and use of the ships magnetic compasses.

**Keywords:** magnetic compass; compass error; residual deviation; magnetic compass compensation

## 1. Introduction

The introduction of contemporary technologies in design and marine equipment has led to significant improvement and development in maritime navigation, including the development of unmanned and autonomous ships and e-navigation. The most recent required system is the Electronic Chart Display and Information System (ECDIS). The use of ECDIS with the Global Navigation Satellite System (GNSS) and other mandatory navigation devices enable the permanent display of a ship's position and heading and other relevant navigational information. Apparently, the problem of determining and continuously displaying the ship's position at any location in the world's oceans has been solved.

Keeping the vessel on a given course is usually carried out with the aid of a gyrocompass [1], which is the master compass on board. The reliability of the compass plays a central role in any steering mode. Its accuracy is much greater than that of magnetic compasses; however, even so the gyrocompass is still not accurate enough to ensure safe berthing of large ships in many ports with narrow basins and restricted approaches. In fact, in practice, the gyrocompass often does not meet the standards established by the International Maritime Organization (IMO) [2]. Late or missed detection of a gyrocompass malfunction can lead to accidents at sea. In recent years, there have been some occasions when a sudden, unexpected loss of power triggered the undetected inaccuracy of electronic instruments, which then developed into a serious crisis. To support this contention, one need only recall the 2014 accident involving the m/v "Atlantic Erie", a bulk carrier

which ran aground in Ontario. The crew was unaware of a gyrocompass defect. If they had the ability and readiness to switch to "old fashioned" navigation using a reliable magnetic compass, that should have prevented the grounding [3].

The magnetic compass's advantage is entirely related to its simplicity, durability, and autonomy [4]. A transmitting magnetic compass with integrated correction of magnetic variation and deviation curve can be used as a source of heading information. Other heading sensors are the free-directional gyro updated by a Kalman filter, rate-of-turn gyro, GNSS compass filtered and stabilized by the inertia of the free-directional gyro, and a vessel's course-over-ground (CoG) received from differential global positioning system (DGPS). With these aids to navigation, the officer of the watch (OOW) has in his hands many reliable sources to identify ships' heading and CoG, but the only independent one—is the magnetic compass.

As with all technological developments, an adjustment period has followed that indicates that the improvements bring with them new maritime navigation problems, such as over-reliance on satellite and computer-based navigation systems and their vulnerability to cyber threats. Today's global maritime sector depends on digitalization, integration of operations, and automation [5]. While automation offers excellent benefits, it also introduces a set of corresponding cybersecurity-related risks [6,7]. According to the International Convention for Safety of Life at Sea (SOLAS) regulation V/19.2.1.1, all ships, regardless of size, shall have a properly adjusted standard magnetic compass, or other means, independent of any power supply, to determine the ship's heading and display the reading at the main steering position [8]. The standard magnetic compass (and spare magnetic compass) must be appropriately compensated, and its table or curve of residual deviation must be available on board in the vicinity of the compass at all times [9].

The IMO also requires that masters and officers in charge of navigational watch know, understand, and have proficiency in regard to the principle of operation and error determination of the magnetic compass [10]. The usual method used on board merchant vessels is comparison between bearing, measured by magnetic compass, and the azimuth of a celestial body. Another method uses bearing to a distant object on shore in the vicinity and comparison in the same way. Thus, despite the view that the magnetic compass belongs to the past of navigation, it retains value as a backup instrument [11]. The International Convention on Standards of Training, Certification and Watchkeeping for Seafarers (STCW) requires that the officer in charge of the navigational watch shall conduct regular checks to ensure that the standard compass error is determined at least once a watch and, when possible, after any significant alteration of course, and that the standard and gyrocompasses are frequently compared and repeaters are synchronized with their master compasses [10]. According to a procedure in the ship's safety management system, it is mandatory to record this in the compass observation (deviation) book [12]. Masters and officers of the watch do not have in their hands options to mitigate compass deviation at the time of the ship's voyage. According to the STCW Convention standards, they do not have such obligations. It is one of the reasons why they must check compasses and record their errors. Noncompliance with these obligations is one reason for remarks in port state control (PSC) inspections. Clearly, the mandates regarding the magnetic compass are no longer being taken quite seriously. The analysis of the annual reports of the regional memorandum of understanding (MoU) on PSC shows that deficiencies related to SOLAS Convention Chapter V (Safety of Navigation) are among the five main reasons for PSC remarks in virtually all regions (ranging from 11 to 15%) [13–16]. Further, the analysis of the top 20 deficiencies (from the Safety of Navigation group) shows extensive compass deficiencies recorded, resulting even in "detention of the ship" [17]. To provide some illustration, problems serious enough to record regarding the magnetic compass include air bubbles, lack of spare magnets, missing bearing devices, rust, and—this is key—irregular inspections by crew members [18].

This article aims to underscore the importance of the magnetic compass and exhort navigation officers to properly use and maintain this historic yet still necessary navigation

aid. Further advances in navigation are required before the magnetic compass can be abandoned. For instance, a truly redundant safety measure is not the addition of a second gyrocompass, comforting as that may be. In a sense, this paper is about securing the advances in navigation. Furthermore, classical methods of magnetic compass compensation are discussed, results of adjustments summarized, and findings on the influence of ship types and age on deviation are shown. This article presents the results of a survey of navigation officers regarding the current use and recommendations provided on means to improve the compensation.

This article consists of four sections. Section 2 provides the literature review, Section 3 describes the methodology and presents the results, and Section 4 provides discussion and conclusions.

## 2. Literature Review

The ship's magnetic compass is a classic navigational instrument that has been thoroughly researched. Noteworthy are the achievements of Airy and Kolong that laid the foundation for the compensation and determination of the residual deviation of a compass onboard a ship with a metal hull. In late 1835, Airy conducted a series of studies, as a result of which he developed a method for compensating and determining the residual deviation of a magnetic compass [19]. The deviation device of modern compasses and the additional soft iron of various shapes attached to the compass allows for the application of this method. Compass adjustment is a job that can take from one to several hours if well organized, experienced, and under favorable conditions. Usually, it takes between two to four hours [20].

Kolong's method requires a more complicated sequence of execution, so it is applicable only if the compass adjuster has the necessary equipment. The method is described in detail by Kozhukov [21], and the author also presents the construction of the device invented by Kolong, called a "deflector". The Kolong method is not widely used because of the need for additional instruments, the time required to prepare the compass on shore, and to perform the maneuver of the ship to determine and compensate for the residual deviation.

Airy's method puts into practice the theoretical developments of Smith and Evans, published in 1863 in their work "Admiralty Manual for Ascertaining and Applying the Deviations of the Compass Caused by the Iron in a Ship" [22]. Evans–Smith's formulae, for which the magnetic course is plotted in the Fourier series, is the basis for the entire mathematical means for calculating the deviation table of any magnetic compass [23]:

$$\tan \delta = \frac{A' + B' \cdot \sin \zeta + C' \cdot \cos \zeta + D' \cdot \sin 2\zeta + E' \cdot \cos 2\zeta}{1 + B' \cdot \cos \zeta - C' \cdot \sin \zeta + D' \cdot \cos 2\zeta - E' \cdot \sin 2\zeta} \tag{1}$$

The symbols $\zeta$ and $\zeta'$ denote the magnetic and compass course, whereas $A'$, $B'$, $C'$, $D'$, and $E'$ indicate the exact coefficients that were expressed originally by Archibald–Smith as $A$, $B$, $C$, $D$, and $E$ [24].

Another formula widely used in practice for determining the residual deviation (2) allows the calculation of the deviation in degrees simply, although various trigonometric estimations turned it into a rough equation [25–27]. This approximate deviation ($\Delta$) is normally used to adjust the magnetic compass [28–30]:

$$\Delta = A + B \cdot \sin \zeta' + C \cdot \cos \zeta' + D \cdot \cos 2\zeta' + E \cdot \cos 2\zeta' \tag{2}$$

The symbols $A$, $B$, $C$, $D$, and $E$ are approximate deviation coefficients and their values match up with the sine of the exact coefficients. Both coefficients may be considered constant for a long time [2,24]. Nevertheless, this may not always be the case, since a bolt of lightning or a shipment of steel cargo may affect the ship's magnetism [31].

After transformations (1), formulas are derived for the exact coefficients of the deviation of the magnetic compass (3):

$$
\begin{aligned}
A &= \frac{\delta_N + \delta_S + \delta_E + \delta_W + \delta_{NE} + \delta_{SW} + \delta_{SE} + \delta_{NW}}{8} \\
B &= \frac{\delta_E - \delta_W}{4} + \frac{\delta_{NE} - \delta_{SW} + \delta_{SE} - \delta_{NW}}{4} \cdot sin45° \\
C &= \frac{\delta_N - \delta_S}{4} \frac{\delta_{NE} - \delta_{SW} + \delta_{SE} - \delta_{NW}}{4} \cdot cos45° \\
D &= \frac{\delta_{NE} + \delta_{SW} - \delta_{SE} - \delta_{NW}}{4} \\
E &= \frac{\delta_N + \delta_S - \delta_E - \delta_W}{4}
\end{aligned}
\tag{3}
$$

It is important to note that magnetic compass deviation calculations start with coefficient calculations. Because of this, the choice of the deviator between Formulae (3) and (4) for coefficient calculations is essential for the accuracy of his work.

The practical application of the Formula (3) requires the ship to steer successively on eight compass courses: 0°, 45°, 90°, 135°, 180°, 225°, 270°, and 315° [32]. The magnetic compass deviation table is computed for every 10° or 15°, replacing the compass course in Formula (1) with the corresponding value of $\zeta$.

If the ship turns 360°, but only the deviation of the main magnetic courses 0°, 90°, 180° and 270° is measured, the abbreviated Formula (4) can be used to calculate the coefficients:

$$
\begin{aligned}
A &= \frac{\delta_N + \delta_S + \delta_E + \delta_W}{4} \\
B &= \frac{\delta_E - \delta_W}{2} \\
C &= \frac{\delta_N - \delta_S}{2} \\
D &= \frac{\delta_{NE} + \delta_{SW} - \delta_{SE} - \delta_{NW}}{4} \\
E &= \frac{\delta_N + \delta_S - \delta_E - \delta_W}{4}
\end{aligned}
\tag{4}
$$

Lushnikov and Pleskacz argue that the method for determining the residual deviation required by the regulations is overly time-consuming and costly [20]. They propose using a more straightforward method based on the compass' directional force (λH). By increasing the coefficient of the guiding force (λ), there will be a reduction in all types of deviation. Thus, one does not have to deal with individual types of magnetic compass deviation before, but with all deviations simultaneously [21]. This method is based on the installation of a standard suspension device that compensates for the semicircular deviation during the initial installation of the compass and increases the directional force at the same time, or simply, after that, it is not necessary to perform compensation. The disadvantage of this method is that the existing magnetic compasses must be reconstructed, while newly built compasses must be equipped with this device, of course, after thorough testing and evaluation in practice.

Kozhukov also proposes abbreviated formulas for calculating the coefficients. His proposal is based on the permissible value for the magnetic compass deviation and sets ±4° as the limit. If the compass deviation is above this value, the full formulae and the corresponding ship maneuvering method should be used [21]. This theory is also found in other studies [24,33,34]. To be widely accepted, these methods need some additional empirical evaluations and comparisons with the "classical" methods of compensation in the cardinal and intercardinal courses.

It is essential to note that the magnetic field of a ship is variable and depends on several factors, such as the sailing area, the type of cargo the ship is carrying, the type of repairs carried out on board, the installation of additional equipment, the age of the ship, [35,36]. Based on these factors, applying one or another type of formula in the calculation process should be done after a critical analysis of the circumstances.

The STCW-78 Convention does not require deck officers and masters to have the knowledge and skills to adjust the magnetic compass. Deck officers and masters must be able to use the information on magnetic compass readings. Compensation and determination of residual compass error is the responsibility of persons certified by the competent maritime authorities [37]. Attention should be paid to the maritime authorities' provi-

sions, which also allow masters to compensate the deviation and compile a new table with the values of the residual deviation [38–41]. In such a manner, the master needs to be aware that any changes to the ships' hull, to loading/discharging equipment, to machinery, and other mechanisms that lead to changes in the ship's magnetic field and magnetic compass deviation.

Another serious problem affecting the accuracy of the magnetic compass is the latitude error caused by a significant change in the ship's latitude [24,36], especially near the magnetic poles [31]. Any ship's trading operation requires it to travel thousands of miles across different latitudes. The variable nature of the Earth's magnetic field causes the ship's magnetic field to change due to the intersection of the Earth's magnetic lines of force at different angles, as illustrated in Figure 1a. That accumulates an error, as per Figure 1b, in the deviation table, and eventually, the actual deviation of the magnetic compass does not correspond with the deviation table data.

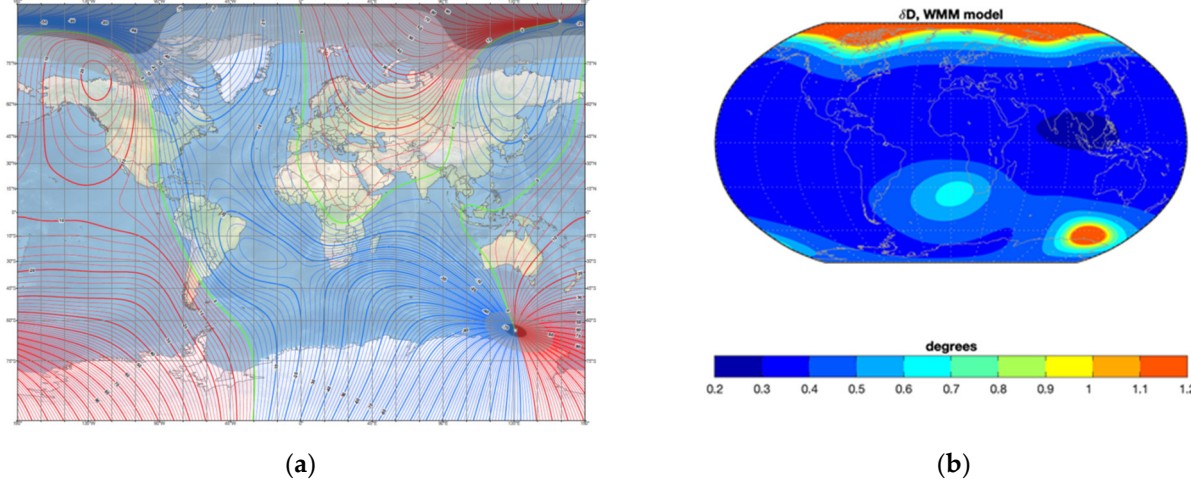

(a)      (b)

**Figure 1.** (**a**) World magnetic model—main field declination [42]; (**b**) global distribution of the declination error model. Color scale limited to 1.2 degrees [43]. Available online: https://www.ngdc.noaa.gov/geomag/WMM/image.shtml (accessed on 27 January 2021).

In this context, studies are being conducted to apply the least-squares method to determine the deviation of any ship's course [44], although the author has not yet tested the algorithm in practice. Felski [45] has proposed a device that automatically collects information about the deviation and continuously compares it with the deviation table values. Other authors propose an algorithm to compute the latitude error while the ship is in service somewhere in the world [33]. However, all the listed improvements in compass adjustment activity are based on the ship's existing deviation table. That means that at least once the residual error's compensation and determination must be performed by a highly qualified and certified compass adjuster.

## 3. Methodology and Results

The study was conducted using two approaches: by analyzing the results of magnetic compass compensation and surveying the current use of compasses on ships.

The compensation of the deviation of a ship's magnetic compass and the subsequent determination of the residual deviation were carried out according to Airy classical method's rules with a full swing of the ship on eight magnetic courses by in situ measurements. The method for determination used by the authors is applicable in the area at sea specially designated for magnetic compass adjustment or in an area with sufficient space for maneuver and accurately distinct onshore markers. Using the gyrocompass course as reference direction is incorrect because of gyro inertia.

Deviation activities begin with turning the vessel on every cardinal course: 000°–180° and 090°–270°. If the compass deviation is within the limit, the vessel is turned to the quadrantal courses: 045°–225° and 135°–315°. If the deviation is within the limit, one full circle from 000° to 360° is necessary to maintain the ship on every cardinal and quadrantal course for at least one minute. The deviation activities are correct if the weather and sea are calm; otherwise, the ship's deviation is affected by rolling and pitching.

For this study, ships were grouped by type and age (years of service). The plots show calculated deviation values based on Formulae (3) and (4) for the selected ship types and ages. Deviation and specific differences that may indicate the need to perform compass compensation using the Airy method were noted.

The second part of the research methodology in this article was a survey. The objectives (targets) of this survey are to review the level of knowledge and current use of magnetic compasses on board. The research instrument was a questionnaire consisting of an introduction, general questions, and specific questions. The introductory part of the questionnaire contains general remarks (i.e., research objectives, instructions for respondents, importance of the survey, and a statement about voluntary and anonymous participation in the survey). The questionnaire contains nine general and specific questions. The questions were closed-ended with predefined single-choice responses. The general questions aimed to categorize the profile of the respondents. These included questions on certificates of competency (CoC), seafaring experience, and assignments on board. The group of specific questions related to the use and ability to set the magnetic compass. This group of questions was divided into two categories. The first category includes questions related to regular deviation checks and intervals to perform this task, use of the deviation table (curve), and experience with PSC inspections of the magnetic compass. The purpose of these questions was to collect data on knowledge of this specific related topic. The second category includes questions about the intensity of the importance and frequency of deviation control and compensation. This category consisted of numerical values ranging from one (lowest level of importance and the lowest level of frequency of performing the task) to five (highest level of importance and the highest frequency of performing the task). Thus, the relationship between the answers was established. After developing the questionnaire, a test of the survey was conducted by the professors of navigation and experienced experts (retired shipmasters). The questionnaire was then corrected and distributed online to the target group of 320 respondents. The survey was conducted from 9 December 2020 to 9 January 2021. During this period, 123 responses were collected from navigation officers. The target group consisted of deck officers and captains. To verify the results and their clear interpretation, a face-to-face interview or correspondence via email was used for a selected 10% of the respondents according to their CoC and assignment on board.

### 3.1. Results of the Magnetic Compass Compensation Study

The compensation and determination of the residual deviation were carried out in the area of Varna Bay in the western part of the Black Sea with approximate coordinates latitude $\varphi = 43°11.0'$ N and longitude $\lambda = 28°55.0'$ E. The total number of ships included in the study was 252. The distribution by type is shown in Table 1.

**Table 1.** The number of ships included in the study according to type.

| Type of Ships | Number of Ships |
|---|---|
| General Cargo | 83 |
| Tankers | 60 |
| Bulk carrier | 57 |
| Reefer | 25 |
| Offshore supply ship | 13 |
| Container ship | 13 |
| Ro-Ro Cargo | 1 |

One of the theses put forward by the authors is that the type of ship is influences the change in the magnetic field. The transported cargo over time and while the ship is sailing at different latitudes changes the magnetic compass' deviation. Experiments have shown that the loading and unloading activities of metal-bearing cargoes and liquid bulk cargoes affect the compass deviation differently. In addition, the transport of goods over long distances in different latitudes of the world's oceans influences the magnetic compass error. In this context, it is important to reiterate the importance of regularly determining compass errors during the voyage.

Another hypothesis that we support and for which we provide evidence is the dependence of the change in magnetic compass deviation on the ship's age. The accumulation of changes, including design changes, alters the ship's magnetic field, and these changes directly affect the deviation. Therefore, the studied ships are divided into five groups according to their age into deviation activities. The cutoff point is five years, the time for class repairs where structural changes are made if necessary. The percentage of the different ship groups by their age is shown in Table 2.

**Table 2.** Distribution of the examined ships depending on their age.

| Group of Ships (Per Their Age) | % |
|---|---|
| 0–5 | 20 |
| 6–10 | 15 |
| 11–15 | 17 |
| 16–20 | 16 |
| >20 | 32 |

The data processing involves calculating the coefficients of deviation of the magnetic compass of each ship using the Formulae (3) and (4). The empirical values of the deviation were calculated when the ships maneuvered to eight compass courses: $0°$, $45°$, $90°$, $135°$, $180°$, $225°$, $270°$, and $315°$. The results show a graph of the deviation constructed with coefficients from the full formulas and one constructed with coefficients from the short formulas. For the calculations using Formula (4), only the deviations on the four main magnetic courses: $0°$, $90°$, $180°$ and $270°$, are considered. The analyses were also performed considering the ship type.

### 3.1.1. Compasses on Newer Vessels

Barring force majeure, ships five years old and less have not undergone any hull and machinery alterations. Thus, the conditions for constructive preservation of the ship's magnetic field are met. Any alterations would result from the nature of the cargo carried. Figure 2a show the deviation curves of a bulk carrier that was two years old at the time of the deviation activities. The graph shows the deviation curve after calculated coefficients according to Formula (3) (red line), and the deviation curve after calculated coefficients according to Formula (4) (blue line).

Figure 2a shows that the deviation of the magnetic compass is mainly caused by the magnetic field of the solid ship's iron and has a semicircular character. The values are approximately the same with respect to maximum and minimum and are within $\pm 5°$. In this case, it may be necessary to reduce the deviation values, but such measures are not mandatory.

Figure 2b shows graphs of a five-year-old tanker. The lines show the deviation curve after calculated coefficients according to Formula (3) (red line), and the deviation curve after calculated coefficients according to Formula (4) (blue line).

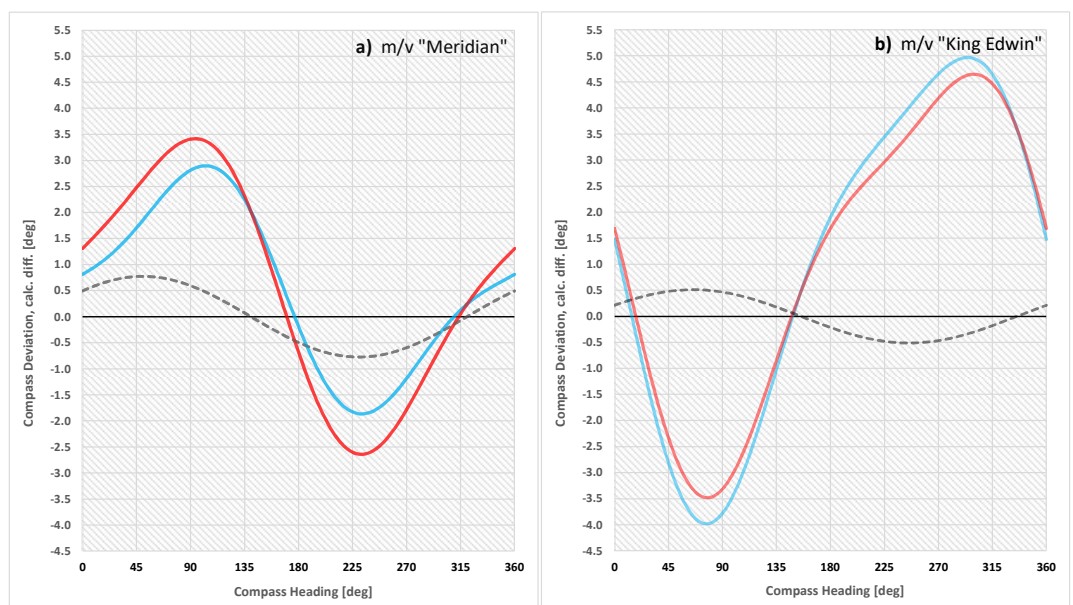

**Figure 2.** Graph of the deviation, according to Formulae (3) and (4): (**a**) m/v "Meridian", (**b**) m/v "King Edwin".

In this case, the same results are observed. The following conclusion can be drawn:

- The type of deviation of a newer ship (age 0 to 5 years) is mainly semicircular;
- Clearly, it takes time for cargo type to affect the magnetism of a vessel—changes are generally a result of latitudinal effects.

### 3.1.2. Compasses on Vessels from Five to Ten Years of Age

The deviations were measured after a class repair of 39 ships 5–10 years old. Figure 3a shows the diagram of a ten-year-old ship carrying petroleum products and chemicals. The magnetic compass deviation is semicircular, but the graphs are shifted along the abscissa. In this case, applying the short formulas to determine only the semicircular deviation leads to a calculated value of coefficient A equal to zero. Although the two figures' deviation is within the allowable values, complete measurements and application of the formulae are necessary.

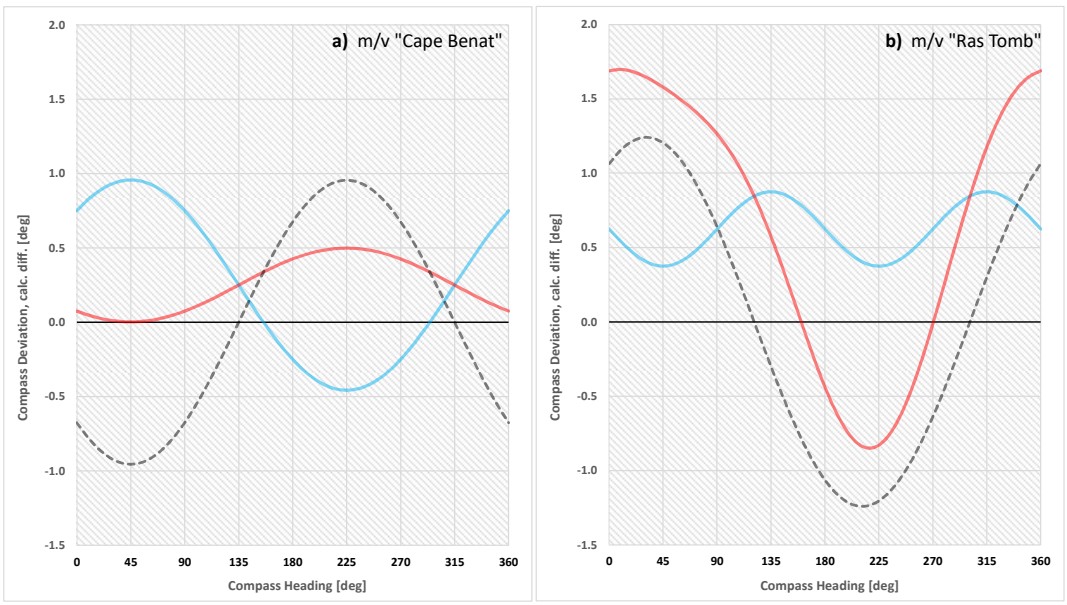

**Figure 3.** Graph of the deviation, according to Formulae (3) and (4): (**a**) m/v "Cape Benat", (**b**) m/v "Ras Tomb".

The other example, in Figure 3b, shows the deviation of a nine-year-old general cargo ship. The lines present the deviation curve as in the previous figures. The ship's sailing area was the Mediterranean, and the North and Baltic Seas—i.e., latitudes no more than 15 degrees above those at which the deviation was determined in 2012. Figure 3b shows that the nature of the deviation is beginning to change to quadrantal. Although the values are still within acceptable limits, this is a sign that ignoring the determination of the deviation of the magnetic quarter courses 0°, 90°, 180°, and 270° will result in erroneous values of the magnetic compass.

### 3.2. Analysis of the Results of the Survey

A survey method was used to collect the data on the use and knowledge level of magnetic compasses. The responses obtained were processed using statistical, descriptive, and comparative methods. Of the total number of respondents, 36 officers (29.3%) hold the CoC for Officer in charge of a navigational watch on ships of 500 GT or more, 20 officers (16.3%) hold the CoC for Chief mate on ships of 3000 GT or more, and 57 officers (46.3%) hold Master mariner unlimited certificates. Other navigation officers who responded to the survey (10 officers) hold the CoC for Master on ships between 500 and 3000 GT, or naval and other certificates issued in accordance with national regulations. In terms of seafarering experience, 24.4% of respondents have 1–4 years of sea service, 25.2% have 4–10 years of sea service, 21.1% have 10–15 years of sea service, and 29.3% have more than 15 years of sea service. The current (or last) assignment onboard ships is the third officer for 16.3% of the respondents, second officer for 28.5%, Chief mate for 23.6%, while 31.7% are Masters. The number, qualifications, seagoing service, and duties of the respondents constitute a relevant sample to draw reasonable conclusions about the knowledge and use of a magnetic compass on ships.

The respondents were asked about conducting a regular deviation check (error determination) of the magnetic compass; 82.9% of the respondents answered positively, while 17.1% answered negatively. A regular deviation check may indicate the need for repair, testing, or adjustment of the magnetic compass [46]. Table 3 presents the response.

**Table 3.** Distribution of responses to the regular deviation check of the magnetic compass.

| Regular Deviation Check of the Magnetic Compass | % |
|---|---|
| At least once a month | 7.3 |
| At least once a week | 11.4 |
| At least once a day | 17.9 |
| At least once a watch | 17.9 |
| At least once a watch and, when possible, after any major alteration of course | 31.7 |
| Other | 13.8 |

Only 49.6% (31.7% + 17.9%; as per Table 3) of the navigation officers perform a regular magnetic compass deviation check following the provisions of the STCW Convention (STCW, 2017, Section A- VIII/2, Part 4-1). Pleskacz obtained almost identical results (53%) in a survey conducted in 2017 among a sample of 212 navigation officers [34]. These responses indicate an inadequate level of knowledge or negligence in applying the relevant provisions of the STCW Convention relating to watchkeeping or the improper use of a magnetic compass on board. Comparing these results suggests that navigation officers generally consider that compass inspection requirements are overly stressed in the STCW.

Clarification was provided, as navigation officers were asked to estimate the importance and frequency of magnetic compass deviation checks. The responses are shown in Figure 4.

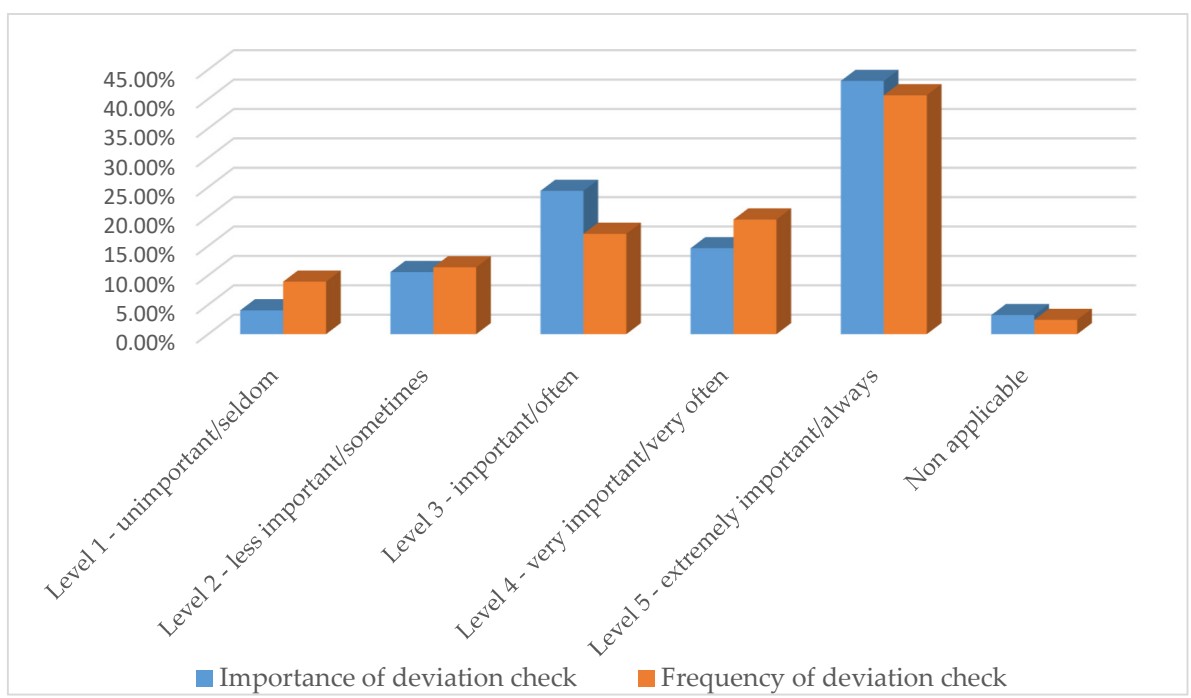

**Figure 4.** The perceived importance and frequency of deviation check of the magnetic compass.

Comparing the data on the regular deviation check (Table 3) with the data on its importance and frequency (Figure 4), we can see that an almost identical percentage of navigation officers perform the regular deviation check of the magnetic compass and at the same time give it the highest level of importance (43.1%) and frequency (40.7%). That suggests that there is disagreement with the required procedures, or a lack of knowledge.

Another critical factor affecting the use of a magnetic compass is the validity of the deviation table (curve). When asked about the significance of the deviation table (curve) in navigation, only 55.3% of the respondents answered positively. This relatively high percentage of responses further suggests the failure of navigation officers to implement regulations regarding the magnetic compass.

The authors compared the responses regarding the regular deviation check and the use of the magnetic compass deviation table with the respondents' current (or last) assignment. The results presented in Table 4 are as expected in that they show equal or very high agreement between the application of STCW regulations and the correct use of the magnetic compass in navigation. Yet the results are somewhat surprising as they show that the highest percentage of responses with positive answers to this question are given by second officers and not by masters and chief mates.

**Table 4.** Responses to the regular deviation check and use of the deviation table by rank.

| Current (Last) Assignment (Nr.) | Deviation Check | | Deviation Table (Curve) Use | |
|---|---|---|---|---|
| | Regular (Nr./%) | Irregular (Nr./%) | Yes (Nr./%) | No (Nr./%) |
| Master (39) | 19/48.7 | 20/51.3 | 19/48.7 | 20/51.3 |
| Chief mate (29) | 12/41.4 | 17/58.6 | 16/55.2 | 13/44.8 |
| Second officer (35) | 22/62.9 | 13/37.1 | 22/62.9 | 13/37.1 |
| Third officer (20) | 8/40.0 | 12/60.0 | 11/55.0 | 9/45.0 |
| Total (123) | 61/49.6 | 62/50.4 | 68/55.3 | 55/44.7 |

The MoU annual reports on PSC show a significant number of shortcomings of the magnetic compasses. Respondents were asked about their experience with PSC inspections

of a magnetic compass—90.2% of respondents answered that their ships had not been subject to PSC remarks regarding magnetic compass deficiencies.

The remaining 9.8% of respondents answered positively. Analysis of the responses revealed that the vast majority (9 out of 13) of deficiencies related to inadequate inspections, significant differences of the magnetic course, or entries in the compass observation book.

## 4. Discussion and Conclusions

Analysis of the magnetic compass compensation results shows that the first-class repair of a ship is the age limit for permanent magnetic field changes. We do not provide examples for the other age groups because the calculations confirm the above conclusion. In 92% of cases, especially for ships older than 20 years, the type of magnetic compass deviation is determined by the type of cargo carried. Relative stability is observed in tankers, reefers, and offshore supply ships. The cargoes carried by these types of ships are nonmetallic. In almost 100% of cases, the deviation has a semicircular character; that allows the use of an abbreviated procedure for its determination. The graph's expected shift can be along the abscissa in the direction of "+" or "−" within a maximum of $\pm 1°$.

For the other types of ships studied, the experiments with Formulae (3) and (4) revealed significant differences in the deviation curves. In almost 60% of the cases, the type of deviation changed over time to quadrantal. This categorically rules out the possibility of using an abbreviated procedure to compensate and determine the residual deviation.

Figure 5 show the deviation curves of the general cargo vessel "Kalitihi Sea" for 2009 (blue line) and 2011 (red line). The ship was built in 1986. In this case, the tendency to change the type of deviation is evident.

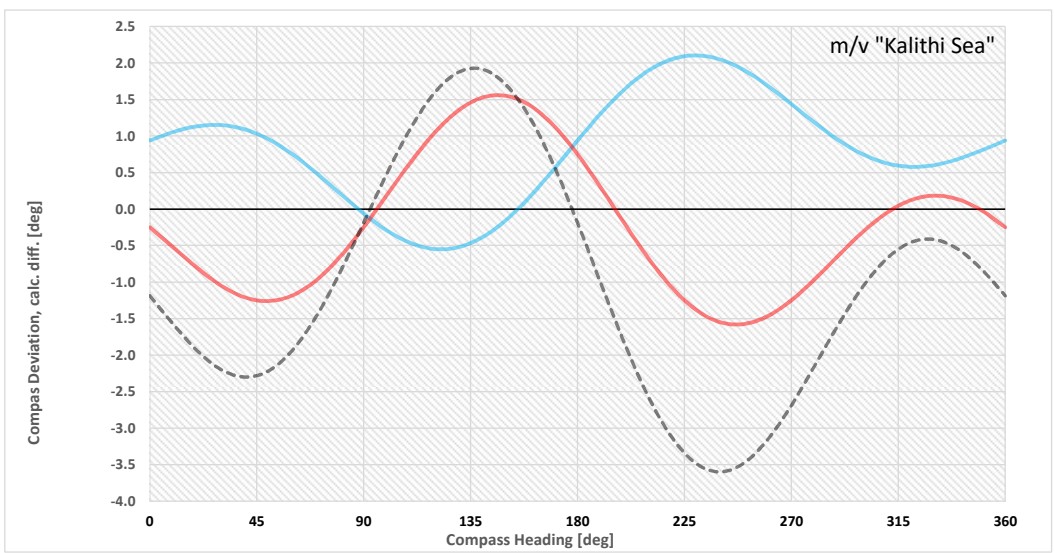

**Figure 5.** Graph of the deviation for 2009 (blue line) and for 2011 (red line), (m/v "Kalitihi Sea").

This and other similar cases suggest that compensation of deviation requires all parties' proper attitudes—ship owners, crew members, flag administrations, PSC inspections, and certified compass adjusters.

Analysis of the methods and techniques used to compensate for and determine the residual deviation of a ship's magnetic compass suggests the following conclusions:

- Ship's masters must instill in all bridge officers a responsible attitude toward magnetic compasses, standard and spare, and maintain them in good working order;
- The adjustment of the standard magnetic compass must be carried out under the requirements of the relevant IMO and flag state regulations;
- Before the commencement of deviation activities, the factors leading to the change in deviation should be analyzed;

- If the analysis shows that there are conditions for the change in the value and nature of the deviation, it is necessary to carry out its compensation by sailing in eight magnetic courses. The calculations should be made using Formula (3) instead of Formula (4) whenever the magnetic compass adjuster is unsure whether the ship's structural change during her life has had significant impact on the magnetic field of the vessel.
- It is important to emphasize that the Airy method is reliable enough and is applicable everywhere worldwide. Another advantage is that the method does not require any special equipment and extra complicated calculations, as required in other methods. The use and application of any innovative technological methods for determining the residual deviation are made based on the previously composed deviation table, in most of the cases calculated by the Airy method.

A regular check of deviation together with the maintenance and application of the deviation table (curve) are the main factors for the proper application of the magnetic compass in navigation. According to the survey results, 14.2% of the respondents said that they do not perform a regular deviation check. In comparing the answers to the question about the periodicities of a regular deviation check, we found that the percentage is significantly higher. Almost 50% of respondents do not perform this activity following the relevant STCW regulations. Analysis of the data on the importance and frequency of the deviation check confirms these findings. The fact that 45.1% of the respondents do not use the deviation chart or deviation table indicates that they do not use the magnetic compass in navigation correctly (if at all). This leads to remarks in PSC inspections where magnetic compass deficiencies are among the top five deficiencies per the analyzed annual reports of regional MoUs.

After analyzing the survey results, the following findings can be elaborated:

- About half of the respondents do not perform regular deviation checks of the magnetic compass, while about 45% of the respondents do not use the deviation table, including masters and chief mates, which is of particular concern;
- Comparing the results of this survey with the 2017 survey conducted by Pleskacz shows a 3.4% decrease in performing the deviation check, which shows how the proper use of a magnetic compass is decreasing;
- The reliability of modern compasses and navigation systems will further reduce the use of magnetic compasses;
- New generations of deck officers and masters will have less need for the use of magnetic compasses;
- These facts will lead to a further reduction in the level of knowledge of the proper use of magnetic compasses, especially in emergencies.

The proper use of a magnetic compass requires navigation officers' joint efforts in regular inspection and maintenance and compass adjusters in regular compensation.

The development of technology in the maritime sector has a significant impact on navigation. This development further reduces the need to use the magnetic compass in navigation. Nevertheless, the magnetic compass remains the only means of indicating course independent of any source of power. The navigation officers need to have the ability and readiness to switch to this "old fashioned" navigation using a reliable magnetic compass in case a gyrocompass malfunction is detected, which could endanger the ship's navigation, safety, and security. Therefore, regular compensation, an adequate level of knowledge, and the correct use of the magnetic compass remain a necessary condition for safe navigation, especially in emergencies.

**Author Contributions:** Conceptualization, A.A., B.B., and I.P.; methodology, A.A., B.B., and I.P.; data collection, B.B. and I.P.; validation, A.A. and M.P.; formal analysis, A.A., B.B., and I.P.; data curation, A.A., B.B., and I.P.; writing—original draft preparation, B.B. and I.P.; internal review, M.P. All authors have read and agreed to the published version of the manuscript.

**Funding:** The authors acknowledge the financial support of the Slovenian Research Agency (research core funding No. P2-0394, Modelling and Simulations in Traffic and Maritime Engineering).

**Conflicts of Interest:** The authors declare no conflict of interest.

**Abbreviations**

| | |
|---|---|
| CoC | Certificate of Competency |
| CoG | Course over Ground |
| DGPS | Differential Global Positioning System |
| ECDIS | Electronic Chart Display and Information System |
| GNSS | Global Navigation Satellite System |
| GT | Gross Tonnage |
| IMO | International Maritime Organization |
| MoU | Memorandum of Understanding |
| OOW | Officer of the Watch |
| PSC | Port State Control |
| SOLAS | Safety Of Life At Sea |
| STCW | Standards of Training, Certification, and Watchkeeping |

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
