# Peer review of "Determining Residual Deviation and Analysis of the Current Use of the Magnetic Compass"

_jmse, doi:10.3390/jmse9020204_

Round 1

Reviewer 1 Report

Dear authors, 

My overall recommendation would be minor-to-major revisions, however, the system does not provide this option. 

I prepared my comments in a separate file, attached for your convenience.

With kindest regards

R#

Author Response

Dear Reviewer,

Thank you very much, indeed for your valuable comments and for your time in reviewing our article. Please allow me to provide you with some comments about changes that we've made (in the article are marked with track changes):

  1. In relation to the English used, I would like to mention that this article was proof-read by a professional. Mr Rick Harsh (Master in Fine Arts, in English) is a professional editor, author and has edited hundreds of papers written in the maritime field. He reviewed this paper, and some changes were made before it was submitted. We cannot, of course, presume that any paper is perfect, but if there are any mistakes in this one or awkward sentences, he would be very happy to revise them if they could be specified.
  2. We have taken into account all your comments related to the scientific contribution; therefore, additional explanations are provided (added text is marked with track changes). The article is now presented in a more transparent and evident manner.
  3. In relation to the calculation of deviation coefficients, we have tried to accommodate answers with some added explanation in lines 43-85, 148-150 and 190-192. We sincerely hope that we have successfully clarified that.
  4. Thank you very much, indeed for the proposed exquisite literature of that we incorporated into our article. Unfortunately, we were not in a position to expand the mathematical part of it, since the reviewer 2 recommendation was to even shorten that part of the article.
  5. We have tried to accommodate answers to your comments on methods and deviation changes in lines 72-74, 82-85 and 190-192. We sincerely hope that we have successfully clarified that.
  6. Thank you very much for pointing out that term Seafarers cannot be a substitution for the OOW or the navigational officers. The term was replaced.
  7. Inline, 20 words newly developed (methods) were deleted.
  8. Term heading was additionally explained and clarified in lines 52-57.
  9. As recommended statement regarding resilience on cyber-attacks was deleted (lines 63-64), since it was obvious that it was redundant.
  10. In lines, 72-74 and 417-425 standard methods of deviation check by the Navigation officer is explained.

Once again, thank you for your time and your work in reviewing our article.

Very respectfully,

Andrej Androjna

Reviewer 2 Report

The paper presents an intersting approach that has to be cosidering in the era of shipping digitization. The added value of the paper needs more elaboration. There are 3 differnt areas the compass compensation, use of magnetic compass with deviation and the perception of users on magetic compass. The last in my opinion is the main added value of paper. It is recommended for authors:

-reduce the details on the theory (mathematics of magnetic compass compensation)

presnte if possible some case in the literrature when magnetic compass is used in an emergency situation (i.e. cyber attack) using if possible some literature references.

 In the study presented in the last part of the paper (which is the most important part of the paper) the following items are needed:

  • Explain research targets  and quationaire structure
  • What methodology was followed and if they did a pilot running to test tools used
  • Perform reliability analysis in response (the sampe is adequate)
  • More detailed description of the findings and how they are connected with the rest of the paper.

Author Response

Dear Reviewer,

Thank you very much, indeed for your valuable comments and for your time in reviewing our article. Please allow me to provide you with some comments about changes we've made (in the article are marked with track changes):

  1. In relation to the English used, I would like to mention that this article was proof-read by a professional. Mr Rick Harsh (Master in Fine Arts, in English) is a professional editor, author and has edited hundreds of papers written in the maritime field. He reviewed this paper, and some changes were made before it was submitted. We cannot, of course, presume that any paper is perfect, but if there are any mistakes in this one or awkward sentences, he would be very happy to revise them if they could be specified.
  2. We have taken into account all your comments related to the scientific contribution; therefore, additional explanations are provided (added text is marked with track changes). The article is now presented in a more transparent and evident manner.
  3. In relation to the calculation of deviation coefficients, we have tried to accommodate answers with some added explanation in lines 43-85, 148-150 and 190-192. We sincerely hope that we have successfully clarified that.
  4. Thank you very much, indeed for the proposed exquisite literature of that we incorporated into our article. Unfortunately, we were not in a position to expand the mathematical part of it, since the reviewer 2 recommendation was to even shorten that part of the article.
  5. We have tried to accommodate answers to your comments on methods and deviation changes in lines 72-74, 82-85 and 190-192. We sincerely hope that we have successfully clarified that.
  6. Thank you very much for pointing out that term Seafarers cannot be a substitution for the OOW or the navigational officers. The term was replaced.
  7. Inline, 20 words newly developed (methods) were deleted.
  8. Term heading was additionally explained and clarified in lines 52-57.
  9. As recommended statement regarding resilience on cyber-attacks was deleted (lines 63-64), since it was obvious that it was redundant.
  10. In lines, 72-74 and 417-425 standard methods of deviation check by the Navigation officer is explained.

Once again, thank you for your time and your work in reviewing our article.

Very respectfully,

Andrej Androjna

Round 2

Reviewer 1 Report

Dear authors, dear corresponding author

Thank you for your thorough explanations and answers. I see that you considered all of the remarks in an appropriate way. Therefore, as for me, the manuscript is OK. 

All the best

R#1

Reviewer 2 Report

The authors perfomed most  modifications suggested. I recommed to improve a liitle their survey (i.e. claculation of Crobach'Alfa ) and if possible to discuss a little the generilization of survey's results